# Dialogue Act-Aided Backchannel Prediction Using Multi-Task Learning

**Wencke Liermann[1], Yo-Han Park[2], Yong-Seok Choi[2], Kong Joo Lee[2*]**

[1] Department of Computer Engineering, Chungnam National University
[2] Department of Radio and Information Communications Engineering,
Chungnam National University
wliermann@o.cnu.ac.kr, {happy115012, yseokchoi, kjoolee}@cnu.ac.kr

## Abstract

Produced in the form of small injections such as "Yeah!" or "Uh-Huh" by listeners in a conversation, supportive verbal feedback (i.e., backchanneling) is essential for natural dialogue. Highlighting its tight relation to speaker intent and utterance type, we propose a multi-task learning approach that learns textual representations for the task of backchannel prediction in tandem with dialogue act classification. We demonstrate the effectiveness of our approach by improving the prediction of specific backchannels like "Yeah" or "Really?" by up to 2.0% in F1. Additionally, whereas previous models relied on well-established methods to extract audio features, we further pre-train the audio encoder in a self-supervised fashion using voice activity projection. This leads to additional gains of 1.4% in weighted F1.

## 1 Introduction

The development of a topic constitutes an integral part of any conversation. At any time, some participant, the speaker, is said to be in control of the topic flow. Simultaneously, other participants attend the talk with various degrees of verbal feedback. Such feedback may take the form of simple vocalic sounds with little to no meaning, e.g., "Uh-Huh", or be extended to more stereotypical phrases, e.g., "Wow!", "Seriously?" (Iwasaki, 1997). In the absence of such so-called backchannels, speakers have been found to tell qualitatively worse stories that they cut off abruptly and half-heartedly (Bavelas et al., 2000). Not only do backchannels shape the narrative, but they also help build a feeling of understanding and trust between conversation partners (Huang et al., 2011; Barange et al., 2022). Therefore, it is unsurprising that continuous research efforts are put into equipping dialogue agents with the all-so-important ability of backchannel prediction. Backchannel prediction

---
*Corresponding author.

is the task of identifying time points in an ongoing conversation at which a competent listener would respond with a certain backchannel type. Since the advent of ChatGPT, modern dialogue systems exhibit answer quality levels on par with humans in various professions (OpenAI, 2023). Once equipped with human-like attentive listening capabilities like backchanneling, their application could be extended to social areas such as counseling and elderly care, not to replace humans but to help where the demand exceeds the supply.

The phenomenon of backchanneling is tightly linked to dynamics of turn-taking, i.e., a fundamental human ability to orderly manage speaking opportunities. Listeners do not intend to claim the turn when producing a backchannel. On the contrary, they use them to signal their continued willingness to remain in the listener position (Iwasaki, 1997). This willingness certainly depends on what the speaker has said. If the previous statement is descriptive or a personal narrative that the listener is just learning about, they might be more likely to encourage the speaker to continue through general backchannels, e.g., "Uh-Huh". In comparison, if the previous statement holds an opinion or some alleged general fact, the listener could agree using a specific backchannel, e.g., "That's right!", but they might as well have reason to dispute and may want to present their own view. First evidence that such a relation between dialogue acts and backchanneling exists was provided by Morikawa et al. (2022) in a preliminary theoretical investigation.

Building on the theoretical observations above, we propose a multi-task model, BCDA, to simultaneously learn the main task of Backchannel (BC) prediction and the sub-task of Dialogue Act (DA) classification. A model trained on the task of DA classification quickly learns to identify utterance types. We allow the BC prediction model to "eavesdrop on" this knowledge and use it as additional guidance to extract more relevant textual features.

We demonstrate the effectiveness of this approach and compare it to the previously suggested sub-task of sentiment classification. Lastly, we verify that BC prediction models benefit from self-supervised Voice Activity Projection pre-training (Ekstedt and Skantze, 2022) even in the presence of large quantities of labeled task-specific backchannel data.

## 2   Related Work

Regarding the task of backchannel prediction, it has been a long-standing observation that audio features are superior to lexical features (Ruede et al., 2017; Ortega et al., 2020), with some early models avoiding the inclusion of the latter entirely (Hara et al., 2018). In the process, *Mel Frequency Cepstral Coefficients* (MFCCs) have established themselves as a simple yet powerful measure of waveform representation (Adiba et al., 2021; Jang et al., 2021) that is only slowly superseded by hidden representations from pre-trained speech recognition models (Ishii et al., 2021). When later models started to incorporate lexical features, they first needed to handle issues of automatic speech recognition quality (Ortega et al., 2020) and the ramifications of possible time delays in fast-paced real-time dialogues (Adiba et al., 2021). Gradually, additional gains obtainable through lexical features convinced several authors to incorporate naive word embeddings (Ruede et al., 2017; Ortega et al., 2020) or, more recently, pre-trained language models (Ishii et al., 2021; Jang et al., 2021). Reflecting the multi-faceted nature of dialogue, backchannel prediction has been performed successfully in tandem with other tasks, such as sentiment classification (Jang et al., 2021) or turn-taking prediction (Hara et al., 2018; Ishii et al., 2021). While results from Hara et al. (2018) suggest that a model trained jointly on the tasks of turn-taking, filler, and backchannel prediction outperforms its single-task variant, Ishii et al. (2021) find that it is not necessarily turn-taking cues itself that are useful, but instead, the participants' turn-management willingness (e.g., willingness to listen, willingness to speak). So far, no multi-tasking approach employed dialogue acts directly.

## 3   Model

### 3.1   Architecture

We apply a multi-modal approach that exploits audio features in combination with lexical features (see Figure 1). The audio-processing component is

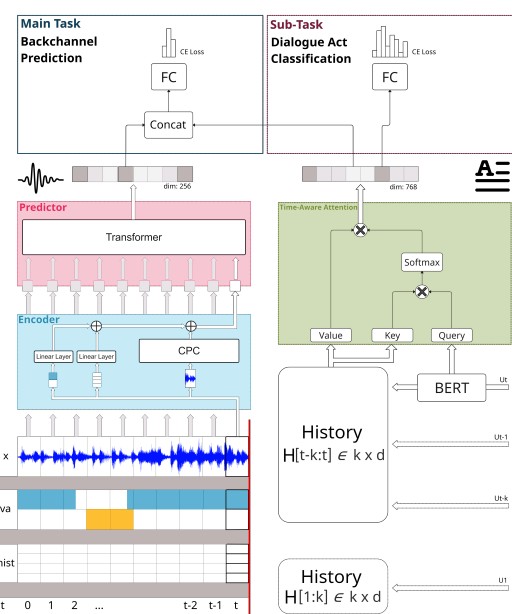

Figure 1: Dialogue Act-Aided Backchannel Prediction (BCDA) model architecture.

taken without alterations from Ekstedt and Skantze (2022). They employ a frame-wise encoder to extract audio features $x_t \in \mathbb{R}^{256}$ at 100 feature vectors (frames) per second. The chosen encoder combines three input types. First, presentations from a frozen speech recognition model (i.e., CPC) (Riviere et al., 2020) are used to encode waveform information. Then, the participants' voice activity, i.e., the notion of whether someone is speaking (1) or not (0) is encoded in a binary vector. Finally, past voice activity is aggregated into a history vector. For two speakers, this aggregation takes the activity ratio of speaker A over speaker B for selected past time intervals. All three feature vectors are mapped to a shared space and added before being pushed through a decoder-only transformer.

As the original authors proposed, we train this architecture using the self-supervised task of Voice Activity Projection (VAP) (Ekstedt and Skantze, 2022). Given an 8s input chunk, the model has to predict a 2s future window for each frame, which is modeled as four bins per speaker of sizes 200, 400, 600, and 800ms. If a speaker is speaking for more than half of the bin, it is marked active (1), else inactive (0). The resulting binary configuration is translated to its decimal counterpart and encoded as a one-hot gold prediction vector. A model pretrained using VAP has been successfully applied to predict turn-taking events, including BC, in zero-shot fashion (Ekstedt and Skantze, 2022). But, it

has not been verified whether VAP can also be a useful initialization step prior to task-specific finetuning. Note that we use pooled speaker & listener audio during pre-training while finetuning and evaluation are performed using speaker-only audio. In this way, we want to ensure that the model is unable to cheat (i.e., pick up on a backchannel in the audio) even if BC timestamps are slightly off.

For the text-processing component, we employ an approach inspired by Malhotra et al. (2022). Given a history of past speech segments, encoded using BERT (Devlin et al., 2019), the segment that directly precedes the current timestamp is projected as the query vector, while the full history serves as keys and values. To reflect the relative importance of a segment in relation to its distance from the current timestamp, the dot product between query and key is scaled down by a monotonically decreasing function of time (i.e., Time-Aware Attention) before the resulting scalars are used to produce a history enriched representation of the current segment. Intuitively, while not all prior speech segments may be directly related to the current one, some may prove valuable to supplement omitted or abbreviated content, i.e., pronoun resolution.

Finally, the representations of both modalities are concatenated and fed through a fully connected softmax layer, resulting in a probability distribution across BC classes. Weight updates are obtained using a cross-entropy (CE) loss.

### 3.2 Multi-task Learning (MTL) Setting

In the MTL setting, the lexical representation is further passed to another fully connected softmax layer to predict the DA of the current segment (if an annotation is available). A probability for each DA is returned and the CE loss is calculated. The total loss for training the multi-task model is a linear combination of the previously mentioned main task loss and this new sub-task loss: $(1 - \lambda) * main\_loss + \lambda * sub\_loss$. Through an exhaustive search, $\lambda$ was set to 0.3

## 4 Experiment

### 4.1 Backchannel Data

In our experiments, we employ the Switchboard Corpus, a corpus of about 260 hours of dyadic small-talk telephone conversations (Godfrey and Holliman, 1993). Based on an aggregated list of surface forms, Ruede et al. (2017) automatically assigned utterances the label BC if they looked

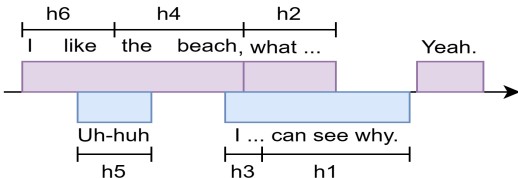

Figure 2: Segmentation of utterances into history segments (h1∼h6) given the backchannel "yeah". h1 is also called the current segment.

like a BC and were a speaker's only voice activity in a fixed interval. To obtain an equal number of negative examples, they defined the timestamp 2s before each BC to be an instance of NO-BC. Later, Ortega et al. (2020) divided their list into generic forms, e.g., "Uh-Huh", and specific forms, e.g., "Wow!" which they assigned the labels Continuer and Assessment, respectively. The result are 122k data instances: 50% NO-BC, 22.5% Continuer, and 27.5% Assessment.

While DA annotations for half the conversations in the corpus are theoretically available (Jurafsky et al., 1997), this annotation was performed on an early version of the transcript before it was resegmented, corrected, and enriched with word-level start- and end-timestamps (Deshmukh et al., 1998). We align the two transcript versions using a naive Levenshtein distance-based approach. Despite an imperfect alignment and the fact that only 48.2% of all conversations have DA annotations, to begin with, we manage to obtain labels for 41.9% of all utterances preceding a BC (or NO-BC). Out of a total of 42 DA tags, the most frequent ones are statement-non-opinion/non-disputable (68.5%) and statement-opinion/disputable (24.2%).

### 4.2 Preprocessing

Each BC instance is linked to a timestamp, and the 1.5s of audio preceding it will be used as input to the model. Additionally, we need to represent textual transcriptions as a history of speech segments. To handle utterance overlap and interruptions, we propose the following segmentation. Given the timestamp of an instance, we collect all the preceding words, sorting them by their start time. We then aggregate neighboring words into one segment if they originally belonged to the same utterance (this implies them being pronounced by the same speaker). An example is given in Figure 2. The utterance "I can see why" is split into two segments to accommodate the interjection "what", which it-

| Model | Trainbl. Params. Audio | Text | NO-BC | Continuer | Assessment | Weighted F1 |
|---|---|---|---|---|---|---|
| **Baselines** (* reimplementation) | | | | | | |
| Ortega (Ortega et al., 2020) | 2.3K | 24.0K | 72.4 | 41.6 | 47.0 | 58.4 |
| BPM_ST* (Jang et al., 2021) | 146.0K | 109.0M | 79.6 | 41.5 | 50.3 | 63.0 |
| BPM_MT* (Jang et al., 2021) | 146.0K | 109.05M | 79.7 (+0.1) | 41.8 (+0.3) | 50.1 (-0.2) | 63.0 (+0.0) |
| **Ours** | | | | | | |
| BCDA_ST | 2.6M | 111.0M | 82.5 | **46.8** | 49.4 | 65.4 |
| BCDA_MT | 2.6M | 111.03M | **82.8 (+0.3)** | 46.7 (-0.1) | **51.4 (+2.0)** | **66.0 (+0.6)** |

Table 1: Backchannel prediction results. With the exception of Ortega et al. (2020) shown values are averages across 10 random seeds. Numbers in brackets indicate differences between single-task (ST) and multi-task (MT) models.

| A | PT | T | MT | NO-BC | Continuer | Assessment | Weighted F1 |
|---|---|---|---|---|---|---|---|
| + | - | - | - | 79.77 (±0.05) | 43.62 (±1.60) | 42.40 (±1.55) | 61.36 (±0.25) |
| + | + | - | - | **80.80 (±0.16)\*** | 44.03 (±1.09) | **45.11 (±0.90)\*** | **62.71 (±0.15)\*** |
| - | - | + | - | 72.79 (±0.47) | 40.52 (±0.82) | 45.46 (±0.66) | 58.00 (±0.37) |
| - | - | + | + | **73.15 (±0.20)** | 40.60 (±0.97) | **46.52 (±0.70)\*** | **58.49 (±0.16)\*** |
| + | + | + | - | 82.52 (±0.16) | **46.77 (±0.69)** | 49.38 (±0.80) | 65.35 (±0.16) |
| + | + | + | + | **82.81 (±0.16)\*** | 46.70 (±0.66) | **51.35 (±0.50)\*** | **66.02 (±0.13)\*** |

Table 2: Ablation study results as averaged across 10 random seeds. Numbers in brackets show one standard deviation. Asterisks mark values that are significantly better than other values in the same block (paired t-test, $p <$ 0.01). (A) Audio encoder. (PT) VAP pre-training. (T) Text encoder. (MT) Multi-task learning using DA prediction.

self becomes one segment. The average number of words in the current segment is 13.4, aligning with related work that used 5 to 20 words as input.

### 4.3 Training Setting

We apply a train-test-valid split suggested by Ruede et al. (2017). The model is trained using an Adam optimizer with a learning rate (lr) of 2e-5 for BERT and the audio component and an lr of 1e-4 for everything else. The former uses a warmup scheduler that gradually increases the lr over 10% of training time before decreasing it again to zero. The latter uses a linear decreasing lr scheduler. During loss computation, we assign each class a rescaling weight (NO-BC: 0.8, Continuer: 1.3, Assessment: 1.1) to counteract class imbalance. The chosen history size is 3, batch size 16, and the number of training epochs 10. The best model is saved based on the weighted F1 of the validation set. Each model configuration was trained on ten different randomly chosen seed values and test set performance values are averaged across those.

## 5 Results

We compare our results to the models of Ortega et al. (2020) and Jang et al. (2021). Jang et al. (2021) perform backchannel prediction in tandem with sentiment classification. Initially proposed for private Korean counseling data, we reimplement their approach using labels from a pre-trained English sentiment classifier. Surprisingly, we ob-

served no significant gains when the sub-task of sentiment classification was introduced. Notably, our model outperforms all baselines by at least a margin of 3% in weighted F1. This performance increase comes at the cost of a comparably small increase in model size of about 4.5M parameters compared to BPM. In comparison, BPM replaced the traditional word2vec embedding employed by Ortega with the powerful BERT model. Despite an additional 109M parameters, they outperformed Ortega by no more than 4.6% in weighted F1.

To provide insight into the origins of our improvements, we perform an extensive ablation study (see Table 2). As illustrated, an audio-only model outperforms the much heavier text-only model by more than 3.3% in weighted F1. However, both components have complementary strengths. While the audio component excels at predicting CONTINUER, the text component shows better results in predicting ASSESSMENT. Their combination outperforms both single modality models by at least 4.0% in weighted F1. This shows the value of including text information.

Additionally, we performed significance tests for each pair of rows in Table 2, evaluating differences between a naive audio-only model vs. a pre-trained audio-only model, a text-only single task model vs. a text-only multi-task model, and a full single task model vs. a full multi-task model. First of all, VAP pre-training leads to a significant performance increase across two out of three categories,

with an overall increase in weighted F1 of around 1.4%. This proves the effectiveness of this form of pre-training, even in the presence of large volumes of labeled data. Finally, it can be observed that the proposed sub-task of dialogue act prediction benefits mainly and to large extent the performance on ASSESSMENT (+2.0%), with only minor but significant gains on NO-BC (+0.3%) and no significant effect on CONTINUER. This might be explained by the fact that ASSESSMENT holds specific backchannel forms, e.g., "Wow.", which are very much content-dependent. Therefore, a better encoding of the context through content-based multi-task learning can help in correctly predicting them. In contrast, CONTINUER are generic backchannel, e.g., "Uh", which can often be inserted arbitrarily and independent of content.

## 6  Discussion

When applied to counseling data, the sub-task of sentiment classification has been previously shown to yield considerable improvements of as much as 3% in weighted F1 (Jang et al., 2021). The discrepancy between those findings and ours could be explained by a difference in the nature of the data. While counseling data is emotion-loaded and the counselor committed to validating those, casual conversations between strangers are quite the opposite. In our data, more than 72.6% of all backchannels have been produced in response to a completely neutral utterance. The proposed sub-task of dialogue act classification constitutes a promising extension and alternative, especially in such casual conversations where the sub-task of sentiment classification fails. Our dialogue act-aided multi-task model achieves an accuracy of 75.12% at the task of dialogue act classification. It is especially good at the differentiation between statement-opinion (sv) and statement-non-opinion (sd). Jurafsky et al. (1997) describe sv as viewpoints that the listener could conceivably have their own (possibly differing) opinion about, while sd are non-disputable descriptives or narratives. They further state that this distinction was introduced to reflect "the different kinds of responses [they] saw to opinions (which are often countered or disagreed with via further opinions) and to statements (which more often get [...] backchannels)" (Jurafsky et al., 1997, Section 5.1.1). While they express doubts about whether or not this distinction is actually fruitful, we have shown that it provides indeed

useful hints for the prediction of certain types of backchannels.

## 7  Conclusion

We proposed a multi-task learning scheme that effectively incorporates dialogue act annotations to enhance backchannel prediction without establishing any requirement for such annotations to be available at inference time. We demonstrated its superiority over the previously suggested sub-task of sentiment classification in casual settings void of strong emotions. Especially, the prediction of specific backchannels could be improved by as much as 2%. Moreover, we have shown that self-supervised pre-training using voice activity projection benefits the performance of backchannel prediction models. Overall, we outperform previous baselines by 3%. All experiment code is made publicly available[1].

## Limitations

In comparison to simple intuitive tasks like sentiment analysis that can be performed by any layperson, dialogue act annotation is commonly performed by linguists, researchers, or other experts in discourse analysis. While it is possible to train a layperson to perform this task with proper guidelines and instructions, the quality of the resulting annotation likely varies. Experts, on the other hand, are often expensive to hire. As for any other task, automatic classifiers exist, but they are still insufficiently accurate, e.g., 83.2% accuracy on the Switchboard Dialog Act Corpus (He et al., 2021).

Moreover, researchers have proposed dialogue act label sets of varying granularity and focus. Each such set might be less or more beneficial when used in tandem with backchannel prediction. The label set applied in this study might not be representative of every possible label set, just as it might not be among the most effective ones.

## Acknowledgements

This work was supported by Institute of Information & communications Technology Planning & Evaluation (IITP) grant funded by the Korea government(MSIT) (No.RS-2022-00155857, Artificial Intelligence Convergence Innovation Human Resources Development (Chungnam National University))

---

[1]https://github.com/wencke-lm/BCDA

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

# A Baselines

This section provides a more in-depth explanation of the chosen baselines. We selected two baselines from previous literature for comparison.

Ortega et al. (2020) employed CNNs in tandem with word2vec embeddings, MFCC features and a listener embedding. They use the same data as is applied in our study. While they do not directly report F1 values, they provide a confusion matrix based on which those could be calculated.

Jang et al. (2021) employed a pre-trained BERT language model together with an LSTM on top of MFCC features. As no public implementation is available, we reimplemented their model based on the information available in the paper. The only parameter missing from their explanation was the hidden size of the LSTM module. After some experimentation, we set this parameter to 128. In their multi-task approach, the original authors used a Korean sentiment dictionary that assigned each word in a collection one out of five sentiments (very negative, negative, neutral, positive, very positive). To the best of our knowledge, a fine-grained dictionary such as this one is not available in English. Therefore, we experimented with some alternative approaches to include sentiment classification. Given a pre-trained sentiment classifier for English[2], we obtain a probability distribution across the sentiments: negative, neutral, and positive. We can now either predict this distribution directly, using KL divergence for loss propagation or encode the category with the highest probability as a hard label, using cross-entropy for loss propagation. The latter approach yielded better performance values, so we used it in our experiments. Input to the model is the first segment joined with any following segment until the speaker changes or the history size (of 3) is exceeded.

---

[2]https://huggingface.co/cardiffnlp/twitter-roberta-base-sentiment-latest