# OpenReview forum: "Dialogue Act-Aided Backchannel Prediction Using Multi-Task Learning"
_EMNLP/2023/Conference — EMNLP 2023 Findings_

### Official Review · Reviewer_ww4h · 2023-08-01

**Soundness:** 3

**Excitement:**

3: Ambivalent: It has merits (e.g., it reports state-of-the-art results, the idea is nice), but there are key weaknesses (e.g., it describes incremental work), and it can significantly benefit from another round of revision. However, I won't object to accepting it if my co-reviewers champion it.

**Missing References:**

He, Zihao, et al. "Speaker Turn Modeling for Dialogue Act Classification." Findings of the Association for Computational Linguistics: EMNLP 2021. 2021.
Skantze, Gabriel. "Turn-taking in conversational systems and human-robot interaction: a review." Computer Speech & Language 67 (2021): 101178.

**Paper Topic And Main Contributions:**

This paper introduces a multi-task learning approach for backchannel prediction in dialogues, enhancing textual representations and utilizing self-supervised pre-training for audio encoders. Outperforming previous models by 3% in WF1, improving computational understanding of back-channeling for applications like conversational AI and speech recognition.

**Questions For The Authors:**

1. From Table 2, adding the proposed multi-task learning increases merely by 0.6% accuracy (65.5% —> 66.1%); a significant test is needed to prove the effectiveness, such as a single-tail t-test. Most of the improvement comes from using the existing pre-trained audio encoder.
2. In the abstract: "we further pre-train the audio encoder in a self-supervised," the authors actually use the audio pre-trained model to initialize the audio encoder. It seems that the paper does not involve a pre-training stage.
3. Please provide the detailed distribution of the 42 DA classification. There is only 40+1 DA (plus one empty) in the training code, and the statement-non-opinion (68.5%) and statement-opinion (24.2%) is not match the distribution in SwDA [1].
4. Turn-taking is just a subset of dialog act, where in this work, the author frames it as a three-way classification (no-BC, BC-Continuer, BC-Assessment). The author may compare the traditional DA classification method with BCDA_MT. For example, [2] could achieve 83.2% ACC in the 42 DA classification.

[1] https://compprag.christopherpotts.net/swda.html#tags
[2] He, Zihao, et al. "Speaker Turn Modeling for Dialogue Act Classification." Findings of the Association for Computational Linguistics: EMNLP 2021. 2021.

**Reasons To Accept:**

1. Proposed a multi-tasking method that jointly models 3 way classification of and 42-way classification of DA in a speech-text multimodal approach
2. Achieve improvement compared with previous methods.

**Reasons To Reject:**

1. Innovations are some how limited, a combination of existing work. Experimental settings/datasets from Ortega, Multi-task from BPM_MT, and pre-trained audio encoder from Ekstedt (VAT).
2. The improvement from the proposed idea is incremental (<1%), and a significant test is needed to prove its effectiveness.
3. Lack of in-depth analysis and insights.

**Reproducibility:**

4: Could mostly reproduce the results, but there may be some variation because of sample variance or minor variations in their interpretation of the protocol or method.

**Reviewer Confidence:**

4: Quite sure. I tried to check the important points carefully. It's unlikely, though conceivable, that I missed something that should affect my ratings.

**Typos Grammar Style And Presentation Improvements:**

The manuscript is more of a technical report than a research paper. The author should focus on the original part rather than the known technique.

---

> ### Author Rebuttal · Authors · 2023-08-29
>
> First of all, we want to thank you for your in-depth feedback and all helpful suggestions.
> Hereby, we want to provide the requested information.
>
> **Question 1: Is the improvement significant?**
> In order to answer this question we perform two significance tests. First, we perform a paired t-test. As we do not perform cross-validation and have only one fixed test set at our disposal, we apply bootstraping to the test set to obtain a collection of performance values. For this purpose we choose 1000 items from the test set with replacement, we repeat this process 10 times to obtain a total of 10 test sets. We then calculate the weighted F1 for each set and use the obtained paired values (one value for the single task model, one value for the multi task model) to perform a paired t-test. We repeat the whole process 1000 times and average the p-values across those runs. The obtained average p-value is 0.000967 which is smaller than 0.05 and therefore indicates that our results are significant.
>
> **BUT** we do not believe in the truthfulness of this result. One requirement of the t-test and most other significance tests is that the performance values must be independent of each other. As we draw all samples from the same set, samples will overlap and are therefore definitely dependent. A violation of the independence assumption causes a large Type I error, that is, it becomes more likely to claim something as significant which might in fact just be a random difference.
>
> To the best of our knowledge, the only significance test really applicable in our situation is the McNemar-Test. However, this test cannot measure whether a difference in F1 is significant, but only whether the error counts of two models are significantly different. Given a matrix of error counts:
>
> | | BCDA\_MT Correct | BCDA\_MT Wrong |
> | --- | --- | --- |
> | BCDA_ST Correct | 7077 | 471 |
> | BCDA_ST Wrong | 546 | 3276 |
> (Test set size: 11,370)
>
> We perform the McNemar-Test in R and obtain:
> McNemar’s chi-squared = 5.531, df = 1, p-value = **0.01868 (< 0.05 => significant)**
>
> This shows that the multitask model is significantly more often correct than its single task variant.
>
> **Question 2: Does our paper involve a pretraining stage?**
> Indeed it does. The audio encoder first encodes the audio waves into feature vectors. For this purpose we apply an already pretrained speech recognition model. This model remains always frozen and will not be further finetuned. On top of this feature encoder sits a decoder-only transformer which is randomly initialised. We pre-train this transformer using the task of Voice Activity Projection by Ekstedt (2022). It is true that Ekstedt proposed their pretraining for the Switchboard Corpus but we did not take their pretrained model, but instead performed the pre-training ourself. Why? Because Ekstedt uses cross validation, while we have a fixed split that is different from their division into train-valid-test.
> *So what is our contribution?* Ekstedt evaluated their pre-training only with regard to the model's zero-shot performance on the task of backchannel prediction. They never evaluated whether the proposed pre-training objective would bring any additional gains in the presence of labeled task-specific backchannel data that a model can be trained on directly.
>
> **Question 3: Why does the stated distribution of dialogue acts not match the official documentation of the SWDA?**
> We do not let our model predict the dialogue act for any arbitrary utterance. Instead, each item in our backchannel data has a history of past utterances, the most recent one called *current utterance*. Our model predicts only the dialogue act of this current utterance. Therefore, we obtain a different distribution of dialogue acts. All backchannel data items and their history, including the dialogue act of the current utterance can be found in the submitted code at: `data/swb/utterance_is_backchannel_with_da_with_context.csv`
>
> Here is our detailed distribution (across all splits):
> sd: 34998 (68.4879%)
> sv: 12357 (24.1815%)
> qy: 721 (1.4109%)
> %: 437 (0.8552%)
> ^q: 291 (0.5695%)
> qy^d: 258 (0.5049%)
> ad: 209 (0.409%)
> qh: 201 (0.3933%)
> bf: 197 (0.3855%)
> aa: 183 (0.3581%)
> ba: 156 (0.3053%)
> qw: 155 (0.3033%)
> na: 121 (0.2368%)
> h: 94 (0.1839%)
> fc: 91 (0.1781%)
> nd: 83 (0.1624%)
> ng: 80 (0.1566%)
> qo: 58 (0.1135%)
> b: 45 (0.0881%)
> ^h: 45 (0.0881%)
> no: 43 (0.0841%)
> ^2: 41 (0.0802%)
> cc: 29 (0.0568%)
> bc: 28 (0.0548%)
> bh: 25 (0.0489%)
> qrr: 23 (0.045%)
> bk: 19 (0.0372%)
> t1: 17 (0.0333%)
> ar: 16 (0.0313%)
> am: 12 (0.0235%)
> qw^d: 11 (0.0215%)
> ny: 10 (0.0196%)
> br: 10 (0.0196%)
> b^m: 8 (0.0157%)
> fa: 8 (0.0157%)
> nn: 6 (0.0117%)
> t3: 5 (0.0098%)
> fp: 4 (0.0078%)
> bd: 4 (0.0078%)
> ^g: 1 (0.002%)
> ft: 1 (0.002%)
>
> **Question 4: Is our task just a subset of dialogue act classification?**
> No. Backchannel identification is a subset of dialogue act classification, but not our task of backchannel prediction. The difference is that former has the backchannel utterance as well as its surrounding (previous and following) context at its disposal to perform a prediction, while we only have the previous context available.
>
> **Question 5: How good is our model BCDA_MT at the task of dialogue act classification?**
> Our model achieves an accuracy of 75.12% at this task. It is especially good at the differentiation between statement-non-opinion and statement-opinion, as well as the differentiation between several question types.
> While this falls short of the 83.2% achieved by Zihao et al. (2021), readers should not forget that we weighted the dialogue act classification loss by a lambda of 0.3, giving it less importance.
>
> We hope all your questions were answered. Thank you.

---

### Official Review · Reviewer_fpPn · 2023-08-03

**Soundness:** 4

**Excitement:**

3: Ambivalent: It has merits (e.g., it reports state-of-the-art results, the idea is nice), but there are key weaknesses (e.g., it describes incremental work), and it can significantly benefit from another round of revision. However, I won't object to accepting it if my co-reviewers champion it.

**Paper Topic And Main Contributions:**

This paper proposes a multi-modal approach to backchannel prediction, trained using multi-task learning. They posit that the task of backchannel prediction is closely related to dialogue act classification (and broadly, turn-taking) and propose a multi-task loss to learn both tasks. They improve upon the existing state-of-the-art performance on backchannel prediction and demonstrate that dialogue actions/intents are relevant to backchannel prediction.

**Reasons To Accept:**

1. The proposed approach is creative; it goes against the classic thinking that encoding audio is strictly superior to encoding text for backchannel prediction. Additionally, it is linguistically motivated and is the first paper to tie turn-taking and dialogue act recognition with backchannel prediction using multi-task learning.

2. The experimental results are competitive, improving upon existing state-of-the-art performance. There are also thorough ablations, exploring both unimodal and multimodal networks.

**Reasons To Reject:**

1. A large focus of the paper is on multi-task learning, and the relation between dialogue act classification and backchannel prediction. However, the actual improvements from single-task to multi-task learning are somewhat slim: 65.5 to 66.1. It is not clear whether these improvements are by chance (e.g., would improvements hold across multiple seeds, multiple trials, etc.)

2. The task is somewhat narrow, and the paper only focuses on one corpus, Switchboard, from 1993. However, this narrow focus may still be sufficient for a short paper.

**Reproducibility:**

3: Could reproduce the results with some difficulty. The settings of parameters are underspecified or subjectively determined; the training/evaluation data are not widely available.

**Reviewer Confidence:**

2: Willing to defend my evaluation, but it is fairly likely that I missed some details, didn't understand some central points, or can't be sure about the novelty of the work.

---

> ### Author Rebuttal · Authors · 2023-08-29
>
> Thank you for your kind and thorough review.
> We would hereby like to address your qualms regarding our submission.
>
> **Regarding our choice of data:** Backchannel Prediction is most often researched in the context of counselling conversations. Due to valid privacy concerns any data collected in this context is rarely made public. To the best of our knowledge, the Switchboard Corpus is the largest public corpus of its kind. While we do have access to a smaller private dataset, we agreed that our experiments would be most useful to fellow researchers if performed on a public data set where any results could be readily reproduced.
>
> **Performance:** The overall improvement in weighted F1 might seem small, but it is worth pointing out that the performance in the category *Assessment* increased by a whole 1%. Indeed the trend we observed during hyperparameter was that this category benefited most from the Multitasking with improvements of up to +2.9% (with an overall increase in weighted F1 of 0.8%) in certain configurations. Reason might be that Backchannels of the Category *Assessment* cannot be produced in arbitrary places and are more dependent on the context including the previous dialogue acts.
> Nonetheless, we understand your concerns. To verify whether obtained results are likely to have occurred by chance, we perform a significance test.
>
> To the best of our knowledge, the only significance test applicable in our situation is the McNemar-Test. However, this test cannot measure whether a difference in F1 is significant, but only whether the error counts of two models are significantly different. Given a matrix of error counts:
>
> | | BCDA\_MT Correct | BCDA\_MT Wrong |
> | --- | --- | --- |
> | BCDA_ST Correct | 7077 | 471 |
> | BCDA_ST Wrong | 546 | 3276 |
> (Test set size: 11370)
>
> We perform the McNemar-Test in R and obtain:
> McNemar’s chi-squared = 5.531, df = 1, p-value = **0.01868 (< 0.05 => significant)**
>
> This leads us to believe, that our Multitask Version is significantly more often correct and the observed improvement is very likely not a byprodukt of chance.

---

### Official Review · Reviewer_kE2m · 2023-08-05

**Soundness:** 3

**Excitement:**

3: Ambivalent: It has merits (e.g., it reports state-of-the-art results, the idea is nice), but there are key weaknesses (e.g., it describes incremental work), and it can significantly benefit from another round of revision. However, I won't object to accepting it if my co-reviewers champion it.

**Paper Topic And Main Contributions:**

The authors propose to adopt a multi-task learning paradigm for the backchannel prediction task, with its sub-task being dialogue act classification. This design is motivated by the observation that backchanneling is closely connected to dialogue act, which in turn plays a crucial role in turn-taking prediction. Through their study, the authors effectively demonstrate the efficacy of their approach and provide a comparison to the previously suggested sub-task of sentiment classification.

**Questions For The Authors:**

See reasons to reject.

**Reasons To Accept:**

1. The paper presents a novel approach to backchannel prediction by incorporating dialogue act annotations through multi-task learning.
2. Extensive experiments and ablation studies are conducted.
3. Well-structured and clearly explains the methodology and experimental setup.

**Reasons To Reject:**

1. Lack of model complexity analysis: Considering the marginal performance improvement compared to the baseline and the additional parameters and computational burden introduced by multi-task learning, it is essential for the authors to conduct a model complexity analysis. This analysis will help clarify the potential applications and prospects of the proposed method.

2. The improvement of MT over ST is very limited. The authors should perform significance tests to ensure the reliability of the experimental results.

**Reproducibility:**

4: Could mostly reproduce the results, but there may be some variation because of sample variance or minor variations in their interpretation of the protocol or method.

**Reviewer Confidence:**

4: Quite sure. I tried to check the important points carefully. It's unlikely, though conceivable, that I missed something that should affect my ratings.

---

> ### Author Rebuttal · Authors · 2023-08-29
>
> First of all, thank you for taking the time to review our submission.
>
> ### Model Complexity
> As pointed out, the given short paper did not include a model complexity analysis. To make up for that, we accumulated relevant information on the type and number of all parameters of each compared model in the table below. (as obtained by pytorch_lightning)
>
> | | Audio | Text | Other | Multitask | Total |
> |:---|---|---|---|---| ---: |
> | Ortega (2020) | CNN + Lin. L.: 2.3K | CNN + Lin. L.: 24.0K | Lin. L.: 2.7K | / | 29.0K |
> | BPM (2021)  | LSTM: 146.0K | BERT: 109.0M | Lin. L.: 114.0K | Lin. L.: 49.2K | 109.3M |
> | BCDA (Ours)  | Transformer: 2.6M | BERT + Attention: 111.0M | Lin. L.: 3.1K | Lin. L.: 31.5K | 113.6M |
>
> **Comparison to Related Work Baselines**
> While our model does indeed have the largest number of parameters, it adds only an additional 4.3M compared to the BPM (2021) baseline. This is an approx. 4% increase in model size through which we were able to outperform BPM (2021) by a margin of 3% in F1. In comparison, BPM (2021) replaced the traditional word2vec text embedding employed by Ortega (2020) with the powerful BERT model. This caused the model size to increase by more than 100M parameters, and still they were able to only obtain an increase in F1 of 4.7%. As can be seen, our obtained increase in performance came at the cost of a comparably small increase in model size.
>
> **Comparison Single Task and Multi Task**
> BPM (2021) as well as our suggested model BCDA use a sequence of linear layers on top of the text encoding to implement multitask training. Those linear layers are only active during training and can be entirely removed during inference. Therefore, the multitask approach does not affect inference performance or inference time in any way. However, it is true that during training an additional loss has to be computed and propagated. But the propagation through shallow linear layers is fast and did not noticeable affect training time for either model.
>
>
> ### Significance Test
> Given the quirks of our setup not many significance tests can be safely applied. We do not perform cross validation, and instead use a fixed test set to make our results directly comparable to previous research (Ortega, 2020). While some other authors with similar settings bootstrap the test set to obtain several performance values which could then be used in a t-test or a wilcoxon rank sum test, we deem this as bad practice. This is because, it creates dependent samples and therefore violates the indepence assumption of above tests.
>
> To the best of our knowledge, the only significance test applicable in our situation is the McNemar-Test. However, this test cannot measure whether a difference in F1 is significant, but only whether the error counts of two models are significantly different. Given a matrix of error counts:
>
> | | BCDA\_MT Correct | BCDA\_MT Wrong |
> | --- | --- | --- |
> | BCDA_ST Correct | 7077 | 471 |
> | BCDA_ST Wrong | 546 | 3276 |
>
> We perform the McNemar-Test in R and obtain:
> McNemar’s chi-squared = 5.531, df = 1, p-value = **0.01868 (< 0.05 => significant)**
>
> To conclude, our Multitask Version is significantly more often correct when the Single Task Version is wrong, than the other way around.
>
> We hope we were able to answer all your questions. Thank you.

---

### Meta-Review · Area_Chair_hhRX · 2023-09-20

**Recommendation:** 3

**Metareview:**

This paper proposes an approach to predict backchannels via multi-task training and viewing the problem similarly to dialogue act prediction. The reviewers generally like the novelty of the approach and agree that there are improvements over SOTA, but also comment that those improvements are marginal. The authors reply with additional experiments demonstrating the statistical significance of their results. For these reasons I believe this would be a good candidate for Findings.

---

### Decision · Program_Chairs · 2023-10-07

**Decision:**

Accept-Findings

**Comment:**

This paper proposes an approach to predict backchannels via multi-task training and viewing the problem similarly to dialogue act prediction. The reviewers generally like the novelty of the approach and agree that there are improvements over SOTA, but also comment that those improvements are marginal. The authors reply with additional experiments demonstrating the statistical significance of their results. For these reasons I believe this would be a good candidate for Findings.